# Validation of the Portuguese Adaptation of the Physical Activity and Leisure Motivation Scale (PALMS-p)

**João Lameiras** [1,2,*], **Pedro L. Almeida** [3], **João Oliveira** [1], **Walan Robert da Silva** [4],
**Bruno Martins** [5], **Antonio Hernández Mendo** [6] **and António Fernando Rosado** [7]

1   Portuguese Athletics Federation, 2799-538 Linda-a-Velha, Oeiras, Portugal; vtjoaooliveira@gmail.com
2   GAPP-Psychology and Performance Intervention Group, 2765-605 São João do Estoril, Portugal
3   ISPA-Instituto Universitário, 1100-304 Lisboa, Portugal; pedro@ispa.pt
4   Laboratório de Gênero, Educação, Sexualidade e Corporeidade (LAGESC), Human Movement Science Graduate Studies Program (PPGCMH), Centro de Ciências da Saúde e do Esporte (CEFID), Universidade do Estado de Santa Catarina (UDESC), Santa Catarina 88-035-901, Brazil; walanrobert@hotmail.com
5   GICAFE (Group d'Investigació en Ciències de l'Activitat Física i l'Esport, Universitat de les Illes Baleares), 07122 Palma de Mallorca, Spain; bruno.pt@gmail.com
6   Faculty of Psychology, Universidad de Málaga, 29016 Málaga, Spain; mendo@uma.es
7   Faculty of Human Kinetics, University of Lisbon, 1495-751 Cruz Quebrada, Portugal; arosado@fmh.ulisboa.pt
*   Correspondence: joaolameiras@fpatletismo.pt

**Abstract:** The clear decline in the practice of physical activity (PA) in contemporary society has well-documented problematic consequences in public health. It has led to a clear investment of research efforts in the attempt to identify the psychological constructs associated with health behaviors such as PA, in particular, the motivation that leads people to adopt these behaviors. In this context, the objective of the present study is to present a suggestion of a Portuguese version of the Physical Activity and Leisure Motivation Scale (PALMS), denominated PALMS-p. This instrument evaluates the reasons for the practice of PA. The psychometric qualities of the instrument were evaluated in a sample of 234 participants (86 males, 148 females) who practiced different PA in a recreational context. Confirmatory factorial analysis confirmed the factorial robustness of the PALMS-p ($\chi^2$/df = 2.010 comparative fit index (CFI) = 0.950, goodness of fit index (GFI) = 0.855, Tucker-Lewis Index (TLI) = 0.939 root-mean-square error of approximation (RMSEA) = 0.021, P(RMSEA ≤ 0.05) < 0.001), and the results show that this version presents good internal consistency. The present study corroborates the fidelity and validity of PALMS-p as a motivation measure for the practice of PA in the Portuguese population.

**Keywords:** motivation; physical activity; adaptation; questionnaire

## 1. Introduction

In today's society, the decline in individual adherence to the practice of physical activity (PA) is unequivocal. The literature clearly points out the negative consequences in terms of physical and mental health due to the aforementioned decline [1–3]. Regular PA has been unanimously considered the cheapest instrument of public health, and it is estimated that if physical inactivity reached (only) 50% of the Portuguese population, the costs would be around 900 million euros [4]. Additionally, the practice of regular PA is recognized as having benefits for quality of life, health, and wellbeing fairly described for people of all age groups, namely the improvement of brain health, weight management,

disease reduction, bones and muscles strengthening, and the ability improvement to do everyday activities [5–7].

Regarding the present-day Portuguese reality, the Sports and PA Eurobarometer recently published by the European Commission reveals that Portugal continues to face serious problems concerning the frequent realization of physical exercise or sports [8]. The results of this study also point out that the main reason to practice regular PA is the need to improve health and fitness, and that the more frequently reported barriers are lack of time and lack of motivation.

These conclusions reinforce the need to adapt to the Portuguese population an instrument with psychometric characteristics that allows for a deeper and more precise comprehension of what motivates the practice of PA and, consequently, the potentiation of adherence and general wellbeing of our society.

International research has focused efforts so as to understand individual differences in the adoption of and adherence to PA, striving to identify the motives that lead people to adopt or to not adopt these behaviors, that is, what their motivations are [9–12]. This knowledge of individual motivations to develop a given PA can be decisive for the development of effective interventions with the aim to motivate people to its practice, as well as potentiating adherence to the same [13,14]. With this aim in mind, various instruments have been created and validated to evaluate the motives to practice PA (Table A1) via essentially two types of underlying approaches, theoretical and non-theoretical.

The approach that is supported by theoretical models aims at developing an item and questionnaire structure anchored in the theories of motivation [15]. Despite these instruments being substantiated in a given theory of motivation, they have been insufficient for an encompassing assessment of the broad spectrum of motives to practice PA that have been identified by the research in this scope [16]. Conversely, the non-theoretical approach is based on an empirical exploration of PA participation motives through studies in which researchers identify individual reasons for the practice of PA by means of interviews, develop items based on participants' responses, and determine the underlying factors through statistical methods. However, this approach does not possess a sufficiently solid theoretical framework for the comprehension of this subject matter [15,17].

In an attempt to rectify the limitations mentioned above, Rogers and Morris (2003) created an instrument that resulted from a fusion of the theoretical and non-theoretical approaches, the Recreational Exercise Motivation Measure (REMM) [18]. In the first instance, the authors based themselves on a qualitative approach where through semi-structured interviews they analyzed the reasons for which people partake in non-competitive PA. Subsequently, based on the Self-Determination Theory [19] and an item-to-item analysis, the authors reached a final version of the instrument comprising 73 items grouped into eight factors that define the motives for participation in recreational exercises, namely, mastery, enjoyment, fitness, psychological condition, appearance, expectations of others, affiliation, and competition/ego.

Despite the REMM's unequivocal value, one of the most important limitations attached to it is its great length, which can bring into question the test–retest reliability of the results obtained and, consequently, be inconvenient in its administration in contexts such as sport and exercise [16,20].

Consequently, a more reduced instrument named the Physical Activity and Leisure Motivation Scale (PALMS) was developed through the selection of five items with greater factorial weight in each of the eight REMM factors, originating an instrument composed of 40 items, answered on a 5-point Likert-type scale [20]. This instrument has been adapted in different contexts and realities, namely in Israel [12], Malaysia [15], Iran [21], and Australia [16], revealing a possession of solid psychometric properties concerning its internal coherence and factorial stability.

Given the revised research, the present study aims to translate, adapt, and validate PALMS with a sample of the Portuguese population composed of practitioners of regular PA. It is in this way intended to contribute to investigation carried out in Portugal in regards to comprehension of the motivation to practice PA, a fundamental means of attaining general wellbeing of Portuguese population.

## 2. Materials and Methods

### 2.1. Instrument

The Physical Activity and Leisure Motivation Scale (PALMS) [20] is used to measure participation motives in physical activity, namely, mastery, enjoyment, physical and mental fitness, appearance, expectation of others, affiliation, and competition/ego, being composed of 40 items answered on a 5-point Likert-type scale (1 = totally disagree, 5 = totally agree), with an average completion time of between 10 and 15 min. The score values for each subscale vary between 5 and 25, since each one is composed of five items. In previous research this instrument presented high internal consistency values (mastery $\alpha = 0.78$ to others' expectations $\alpha = 0.82$) as determined using Cronbach's alpha coefficients [12,15,16]. Confirmatory factor analysis indicated a good fit of the applied model of 40 items divided into eight subscales to the data (CMIN/DF = 2.82, NFI = 0.90, comparative fit index (CFI) = 0.91, root-mean-square error of approximation (RMSEA) = 0.06) [15].

According to Molanorouzi et al. (2014), the PALMS criteria validity was supported by the strong significant correlations verified between the different subscales of REMM and those of PALMS ($r_s = 0.86$, $p < 0.001$) [15]. By the same token, the eight PALMS subscales revealed strong test–retest correlations (rs = 0.78 to 0.94), demonstrating the stability of the different measure components over time.

### 2.2. Translation and Adaptation

The translation and adaptation of the original instrument was made through translation–retroversion [22,23]. In the initial stage, the questionnaire was translated to the Portuguese language individually by two bilingual specialists. In a second phase, both translations were subjected to the appreciation of a jury composed of psychologists, trainers, and translators with the aim to compare each of the translated items with their respective originals and choose the ones that best preserved the original meaning and that utilized terms that were more familiar to the Portuguese population (see Table A2). After this procedure, the instrument was named PALMS-p.

### 2.3. Procedures

The participants were first- and second-year students of the courses of Sport and PA and of Social Education in the Human Motricity Faculty, also including participants that developed their PA in several gyms in the Great Lisbon area. The collection of data occurred between October and December of 2016. Participation was voluntary, the participants having declared knowledge of the finality and conditions of execution of the study. The aforesaid questionnaire contained a brief explanation of the study's objectives and the pertinence of the research. The participants were asked to answer sincerely and spontaneously, with clarification that there are no right or wrong answers, and informed of the average questionnaire completion time (around 10 min.)

Lastly, anonymity was assured; that is, in no section of the questionnaire was it required of the participants to identify themselves. Data were also kept confidential, with the data analyzed by the researchers exclusively. Moreover, the participants were informed that at any moment they could ask for additional clarification that they deemed necessary, with access to the general results of the study.

Ethical approval for this investigation was granted by the Faculty of Human Motricity's Ethics Committee.

### 2.4. Data Analysis

Several confirmatory factor analyses (CFAs) were performed using Stata software v. 13 [24] to test the model's adjustments to the data, using the Maximum Likelihood Estimation of Parameters [25]. The presence of outliers was assessed using the Mahalanobis squared distance (D2) and the normal distribution of the data using the asymmetry (Sk) and kurtosis tests (Ku), in their univariate and multivariate tests. The expectation maximization (EM) algorithm was used for absent imputation [26]. The adequacy of the factorial structure to the observed data was assessed using goodness-of-fit indexes:

the chi-square test and the ratio between chi-square and degrees of freedom ($\chi^2$/gL), the comparative fit index (CFI), the Tucker-Lewis index (TLI) and the root-mean-square error of approximation (RMSEA). The chi-square test is an absolute fit index which assumes multivariate normality and is sensitive to sample size [27]. The CFI compares the chi square statistic from the specified model with the chi-square statistic from the null model, in which all variables are uncorrelated. The TLI is an incremental fit index. The RMSEA is an index of the difference between the observed covariance matrix per degree of freedom and the hypothesized covariance matrix which denotes the model [28].

A good chi-square model fit would provide an insignificant result at a 0.05 threshold [29]. Schermelleh-Engel and Moosbrugger (2003) stated that the chi-square ratio indicates good fit when it produces 2 or a smaller value while it indicates an acceptable value when it produces a value of 3 [30]. According to the criteria used (Brown, 2015), CFI and TLI values must be greater than 0.90, preferably above 0.95, and RMSEA values (90% CI) must not be greater than 0.08 [31]. To test the discriminant validity of the factors which is the degree of confidence we have that a trait is well measured by its indicators [32] the extracted average variance index was used, and for internal consistency of the scale, which assesses how reliably the test items that are designed to measure the same construct actually do so, the composite reliability was used [33]. For the extracted average variance index, the Fornell–Larcker criterion [34] was used, which compares the extracted average variance index with the corresponding squared correlation values with other variables. Composite reliability is considered good if more than 0.7, while adequate if more than 0.5 [35].

The parameter invariance technique was used to assess measurement invariance for men versus women and study the adequacy across genders. This technique is used to compare groups of individuals with regard to their level on a trait, or to investigate whether trait-level scores have differential correlates across groups [36]. Cheung and Rensvold (2002) have described the importance of measurement invariance testing between groups in order to determine whether certain measurements can be applied to different groups with different characteristics [37]. Therefore, we performed multi-group analysis between gender.

Invariance assumptions were verified through the differences of CFI ($\Delta$CFI $\leq$ 0.01) in line with [37]. Invariance models were evaluated using several recommendations (e.g., [28]), specifically: for metric invariance, change in Standardized Root Mean Square Residuals ($\Delta$SRMR) of less than 0.030, and change in RMSEA ($\Delta$RMSEA) of less than 0.015 would support model fit; for scalar invariance a change in SRMR ($\Delta$SRMR) of less than 0.010 and change in RMSEA ($\Delta$RMSEA) of less than 0.015 and would indicate good invariance.

In the first unconstrained model (model A), the factor configurations were allowed to differ across groups. When assessing measurement invariance, you begin with the establishment of total configural invariance. In the measurement invariance literature, configural invariance is also commonly referred to as pattern invariance and is considered to be the baseline model. In this level we were only interested in testing whether or not the same items measure our construct across multiple groups. To test this, we estimated both factor models simultaneously. Because this was the baseline model you only need to assess overall model fit to test whether configural invariance holds [37]. In the second model (model B), factor loadings were constrained to be equal to test for factor configuration invariance. In the third model (model C), intercepts were restricted to test for scalar invariance. The ability to justify mean comparisons across groups was established by attaining scalar invariance. Scalar invariance requires that the item intercepts also be equivalent across administrations. Item intercepts are considered the origin or starting value of the scale that your factor is based on. Thus, participants who have the same value on the latent construct should have equal values on the items the construct is based [38].

## 3. Results

*Confirmatory Factor Analysis*

As can be seen in Table A3, when CFA was performed with the original PALMS-p structure (model 1), the model's adjustment indices did not show adequate values. Thus, PALMS-p refinement processes were implemented. In order to make the scale more parsimonious, in the refinement process, items with low factor loading (<0.50) were excluded [36]. Thus, Items 1, 7, 9, 12, 22, and 33 were excluded, so that PALMS-p was composed of 34 items. However, the model's adjustment indicators were still inadequate, showing that the hypothetical measurement model was inconsistent with the observed data; this was interpreted as evidence against the adequacy of the model.

Based on these results, a new refinement of PALMS-p was developed through the modification indices (MI) criterion [31] to assess other sources of model specification. MIs allow for assessing, among other aspects, overlapping of content between the items [31], which is a well-known damage factor to confirmatory factor models [39,40]. Items that showed errors correlated with MI values above 50 were inspected [31]. For each pair of these items, we chose to exclude the one with the lowest factor loading. The pairs of items with MI above 50 were Items 32 and 36 (Item 36 excluded; factorial loading: 0.62); Items 2 and 3 (Item 2 excluded; factorial loading: 0.68); Items 14 and 35 (Item 14 excluded; factorial loading: 0.62); and Items 38 and 30 (Item 30 excluded; factorial loading: 0.62). At the end of this process, Items 36, 2, 14, and 30 were excluded, so that the resulting confirmed PALMS contained 30 items. This 30-item version was evaluated using a new CFA. The results (see Figure A1) presented a more parsimonious version of the scale, with acceptable adjustment rates [$\chi^2$/df = 2.010, CFI = 0.950, GFI = 0.855, TLI = 0.939, RMSEA = 0.021, P(rmsea ≤ 0.05) < 0.001]. These results suggest that the seven-factor model (Competition/Ego; Appearance, Expectation of Others, Affiliation, Physical Condition, Mastery, and Fun) and 30 items should be considered effective.

In addition, all constructs were considered to exhibit discriminant validity, because all average variance extracted (AVE) values exceeded the appropriate square factor correlations (Table A4 and Figure A1). Furthermore, all the factors presented adequate reliability (composite reliability >0.5).

To study the adequacy of the model across genders, the parameter invariance technique was used in order to verify equivalence between the two groups (men and women). As shown in Table A5, the adjustment of the unrestricted model (Model A) (total configuration invariance) was acceptable. The fit of this model provides the baseline value with which all the invariance models specified later were compared. The models with restricted factor loads (Model B) and with restricted intercepts (Model C) (complete scalar invariance) showed a satisfactory fit. The chi-square statistic did not show significant differences between Model A and Model B, and there were also no significant differences between Model A and Model C. Additionally, no differences in the CFI values were found for all model comparisons, so the results demonstrated the models' invariance in both samples, indicating that the factorial structure of the scale was stable between men and women.

## 4. Discussion

In this study, we aimed to translate, adapt, and study the validity in a sample of the Portuguese population of an instrument intended to evaluate the motives for PA: the Physical Activity and Leisure Motivation Scale [20].

The original confirmatory factor analysis revealed that the obtained data of the Portuguese version do not present total overlap with previous versions e.g., [12,15,16,20].

By adopting an exploratory approach, and based on the factorial loadings of the 40 items that compose the original PALMS, six items were eliminated. However, this PALMS-p model was also not confirmed. Therefore, by adjusting the PALMS-p based on modification indices, a model composed of seven dimensions and 30 items was reached. This final version of the PALMS-p revealed a desirable fit between the proposed factorial model and the data obtained from a Portuguese sample of participants in various types of PA. This supports the validity of the PALMS-p construct.

Additionally, the obtained modified model presented convergent and discriminant validity. The achieved results can be considered initial evidence of the convergent validity of factors. In the present study, the Psychological Condition factor was suppressed, and Item 35 "to distract me from other things" that was part of this factor was incorporated into the Fun factor. However, this result is supported by the second-order REMM factors pointed out by Rogers and Morris (2003) in which fun and psychological factors are part of the same dimension (motive/body and mind) [18]. In addition, and despite this fact, REMM and PALMS revealed a high degree of coherence and stability in different populations (see [12,15,16,41,42]), suggesting that the motives for practice of PA pooled into the proposed seven factors are transversal and consequently apply to different cultures and languages.

The factorial structure obtained strengthens the approach anchored in the Theory of Self-Determination [19], the seven factors measured by PALMS-p being possibly categorized as aspects of intrinsic motivation (mastery, enjoyment subscales) and extrinsic motivation (the other five subscales) [11]. According to Molanorouzi and colleagues (2014) these last ones can further be classified into two second-order factors, mind–body motives (physical condition and appearance) and social motives (expectations of others, affiliation, and ego/competition) [15].

Regarding the accuracy of the instrument, composite reliability analysis indicates that PALMS-p is equipped with adequate internal coherence, in line with the values obtained in previous research [12,16]. The scale's totality and the different subscales by which it is composed present composite reliability and average variance extracted values considered acceptable for the dimension of the sample and the scale's number of items [43], which supports the internal coherence and adequacy for evaluating motivations for participation in PA.

Despite the solid psychometric results obtained, the present investigation does present some limitations. The fact that this is a transversal study without test–retest prevents us from making conclusions relating to the stability of the measure and participants' adherence in the long term in their different PA spheres.

In all, the present study presents a proposal of a Portuguese version of a valid and reliable measure that allows for evaluation of the motives to participate in PA (Table A1). Being a potentially important source of information regarding the diverse motives that lead people to participate in different forms of PA, this instrument is intended to contribute to the development of investigation and intervention, for health authorities and for the different parties involved, with the aim of promoting the continued practice of PA with its well-identified and well-documented beneficial effects in regards to physical and mental health.

**Author Contributions:** Conceptualization, J.L. and A.F.R.; data curation, A.F.R., W.R.d.S.; formal analysis, J.L., B.M. and A.F.R.; investigation, J.L. and A.F.R.; methodology, J.L., B.M. and A.F.R.; project administration, A.F.R.; supervision, P.L.A., A.H.M. and A.F.R.; writing—original draft, J.L., W.R.d.S., B.M. and A.F.R.; writing—review and editing, J.L., J.O., W.R.d.S., B.M. and A.F.R. All authors have read and agreed to the published version of the manuscript.

**Funding:** This research received no external funding.

**Conflicts of Interest:** The authors declare no conflict of interest.

## Appendix A

**Table A1.** Evaluation instruments of the motives to practice physical activity (PA) (Pilar Vílchez and De Francisco, 2017).

| Authors | Year | Instrument | Subscales |
|---|---|---|---|
| Gill Gross and Huddleston | 1983 | Participation Motivation Questionnaire (PMQ) or Participation Motivation Inventory (PMI) | Conquest/status, team climate, fitness, energy, development of skills, friendship and enjoyment |
| Gavin | 1992 | Fitness Incentives Quizzes, Body motives | Corporal motives, social motives and psychological motives |

**Table A1.** *Cont.*

| Authors | Year | Instrument | Subscales |
|---|---|---|---|
| Dwyer | 1992 | PMQ Magazine Version | Team orientation, fulfillment/status, aptitude, friendship, development of skills, enjoyment/emotion/challenge |
| Frederick and Ryan | 1993 | Motivation for Physical Activities Measure (MPAM) | Corporal motives, competence, and enjoyment |
| Markland and Hardy | 1993 | Exercise Motivations Inventory (EMI) | Body image and weight, enjoyment, wellbeing, prevention and positive health, competition |
| Markland and Ingledew | 1997 | EMI-2 | Affiliation, muscle endurance and strength, social status, stress control, flexibility and agility, challenges and health emergency |
| Marsh | 1996 | Physical Self-Description Questionnaire (PSDQ) | Physical appearance, muscular strength, endurance, flexibility, health, adherence, excess weight, skills and self-esteem |
| Ryan et al. | 1997 | Motives for Physical Activity Measure—Revised (MPAM-R) | Health, appearance, motor skill and learning of new skills, enjoyment, intrinsic motivation |
| Morris and Rogers | 2003 | Recreational Exercise Motivation Measure (REMM) | Mastery, enjoyment, psychological condition, physical condition, appearance, expectations of others, affiliation, and competition/ego |

**Table A2.** Items and Subscales in the PALMS and its Portuguese translation.

| English Version | | Portuguese Version | |
|---|---|---|---|
| **Subscale** | **Item Wording** | **Subscale** | **Item Wording** |
| Mastery | Improve existing skills<br>Do my personal best<br>Obtain new skills/activities<br>Maintain current skill level | Mestria | Para melhorar as minhas habilidades<br>Para fazer o meu melhor<br>Para desenvolver novas habilidades<br>Para manter o meu nível atual de habilidades |
| Physical Condition | It keeps me healthy<br>It helps maintain a healthy body<br>It helps maintain physical health<br>It improves cardiovascular fitness<br>I will be physically fit | Condição Física | Porque me mantém saudável<br>Porque ajuda a manter o corpo saudável<br>Para manter a saúde física<br>Porque melhora a minha capacidade cardiovascular<br>Porque estarei fisicamente mais em forma |
| Affiliation | Be with friends doing exercise<br>Do activities with others<br>Enjoy spending time with others while<br>Talk with friends while exercising<br>Do something in common with friends | Afiliação | Para estar com amigos enquanto faço exercício<br>Para fazer uma atividade com outras pessoas<br>Porque gosto de passar tempo com outras pessoas<br>Para falar com amigos enquanto fazemos exercício<br>Porque eu gosto de passar o tempo com outras pessoas |
| Psychological Condition | Because it acts as a stress release<br>It's a better way of coping with stress<br>It takes my mind off other things It helps me relax<br>It helps me to get away from pressures | Condição Psicológica | Porque serve para libertar o stresse<br>Porque é a melhor maneira de lidar com o stresse<br>Para me distrair de outras coisas<br>Porque me ajuda a relaxar<br>Porque me ajuda a afastar das pressões |

**Table A2.** *Cont.*

| | English Version | | Portuguese Version |
|---|---|---|---|
| **Subscale** | **Item Wording** | **Subscale** | **Item Wording** |
| Appearance | Improve appearance<br>Improve body shape<br>Define muscles, look better<br>Maintain trim, toned body<br>Lose weight, look better | Aparência | Para melhorar a aparência<br>Para melhorar a forma do corpo<br>Para definir os músculos, melhorar a aparência<br>Para manter o corpo em boa forma e tonificado<br>Para perder peso, melhorar a aparência |
| Others' expectations | It was prescribed by doctor, physiotherapist<br>To manage a medical condition<br>I can earn a living<br>I get paid to do it<br>People tell me I need to | Expectativas dos outros | Porque foi prescrito por um médico, fisioterapeuta...<br>Para controlar um estado clínico<br>Para ganhar a vida<br>Para ser pago para o fazer<br>Porque as pessoas me dizem que preciso |
| Enjoyment | I have a good time<br>It is fun<br>I enjoy exercising<br>It is interesting<br>It makes me happy | Divertimento | Porque é um momento bem passado<br>Porque é divertido<br>Porque gosto de fazer exercício<br>Porque é interessante<br>Porque me faz feliz |
| Competition/Ego | Perform better than others<br>Be more fit than others<br>Work harder than others<br>Be best in the group<br>Compete with others around me | Competição/Ego | Porque tenho melhor desempenho do que os outros<br>Para estar em melhor forma do que outras pessoas<br>Para trabalhar mais do que os outros<br>Para ser o melhor do grupo<br>Para competir com os que estão a minha volta |

**Table A3.** Confirmatory factor analysis of Physical Activity and Leisure Motivation Scale—Portuguese version (PALMS-p).

| Model | $X^2$ (gl) | $X^2$/gl | CFI | TLI | RMSEA (90% IC) |
|---|---|---|---|---|---|
| Model 1 | 1867.892 (753) | 2.48 | 0.451 | 0.478 | 0.178 (0.098–0.234) |
| **Model 2 exclusion from factor loading** | 1146.742 (507) | 2.26 | 0.876 | 0.863 | 0.073 (0.028–0.105) |
| Model 3 final | 773.939 (384) | 2.01 | 0.950 | 0.939 | 0.021 (0.004–0.033) |

**Table A4.** Discriminant validity and composite reliability of the PALMS-p subscales.

| | Item | Coefficient | IC (95%) | Z | *p*-Value | AVE | CR |
|---|---|---|---|---|---|---|---|
| Fun | Palms35 | 0.672 | 0.596–0.748 | 17.35 | 0.001 | 0.63 | 0.91 |
| | Palms37 | 0.830 | 0.783–0.876 | 36.21 | 0.001 | | |
| | Palms13 | 0.837 | 0.797–0.882 | 36.69 | 0.001 | | |
| | Palms25 | 0.841 | 0.797–0.885 | 37.83 | 0.001 | | |
| | Palms3 | 0.766 | 0.707–0.824 | 25.73 | 0.001 | | |
| | Palms34 | 0.821 | 0.774–0.869 | 33.70 | 0.001 | | |
| Appearance | Palms32 | 0.884 | 0.850–0.918 | 51.00 | 0.001 | 0.75 | 0.80 |
| | Palms23 | 0.929 | 0.903–0.956 | 69.14 | 0.001 | | |
| | Palms40 | 0.823 | 0.776–0.870 | 34.43 | 0.001 | | |
| | Palms11 | 0.821 | 0.774–0.868 | 34.20 | 0.001 | | |
| Affiliation | Palms38 | 0.849 | 0.801–0.809 | 34.36 | 0.001 | 0.66 | 0.76 |
| | Palms20 | 0.817 | 0.763–0.872 | 29.54 | 0.001 | | |
| | Palms4 | 0.807 | 0.750–0.864 | 27.97 | 0.001 | | |
| | Palms8 | 0.773 | 0.710–0.835 | 24.22 | 0.001 | | |

**Table A4.** *Cont.*

|  | Item | Coefficient | IC (95%) | Z | *p*-Value | AVE | CR |
|---|---|---|---|---|---|---|---|
|  | Palms29 | 0.825 | 0.772–0.877 | 30.80 | 0.001 |  |  |
|  | Palms17 | 0.806 | 0.751–0.862 | 28.50 | 0.001 |  |  |
| Competition/ego | Palms27 | 0.806 | 0.751–0.862 | 28.25 | 0.001 | 0.67 | 0.83 |
|  | Palms39 | 0.754 | 0.688–0.819 | 22.57 | 0.001 |  |  |
|  | Palms6 | 0.626 | 0.539–0.713 | 14.15 | 0.001 |  |  |
|  | Palms15 | 0.889 | 0.848–0.930 | 42.11 | 0.001 |  |  |
| Physical conditioning | Palms10 | 0.754 | 0.691–0.818 | 23.29 | 0.001 | 0.70 | 0.87 |
|  | Palms28 | 0.855 | 0.809–0.901 | 36.17 | 0.001 |  |  |
|  | Palms21 | 0.620 | 0.502–0.739 | 10.26 | 0.001 |  |  |
| Expectation of others | Palms26 | 0.674 | 0.560–0.789 | 11.58 | 0.001 | 0.50 | 0.70 |
|  | Palms18 | 0.669 | 0.548–0.790 | 10.85 | 0.001 |  |  |
|  | Palms5 | 0.662 | 0.582–0.743 | 16.14 | 0.001 |  |  |
|  | Palms19 | 0.740 | 0.674–0.807 | 21.76 | 0.001 |  |  |
| Mastery | Pams16 | 0.781 | 0.721–0.841 | 25.59 | 0.001 | 0.54 | 0.85 |
|  | Palms24 | 0.801 | 0.744–0.858 | 27.56 | 0.001 |  |  |
|  | Palms31 | 0.679 | 0.601–0.756 | 17.18 | 0.001 |  |  |

**Table A5.** Adequacy of the model across genders.

| Model | $X^2$/gl | CFI | TLI | RMSEA |
|---|---|---|---|---|
| Model A | 1.90 | 0.924 | - | 0.056 |
| Model B | 1.90 | 0.921 | 0.001 | 0.056 |
| Model C | 1.93 | 0.919 | 0.01 | 0.048 |

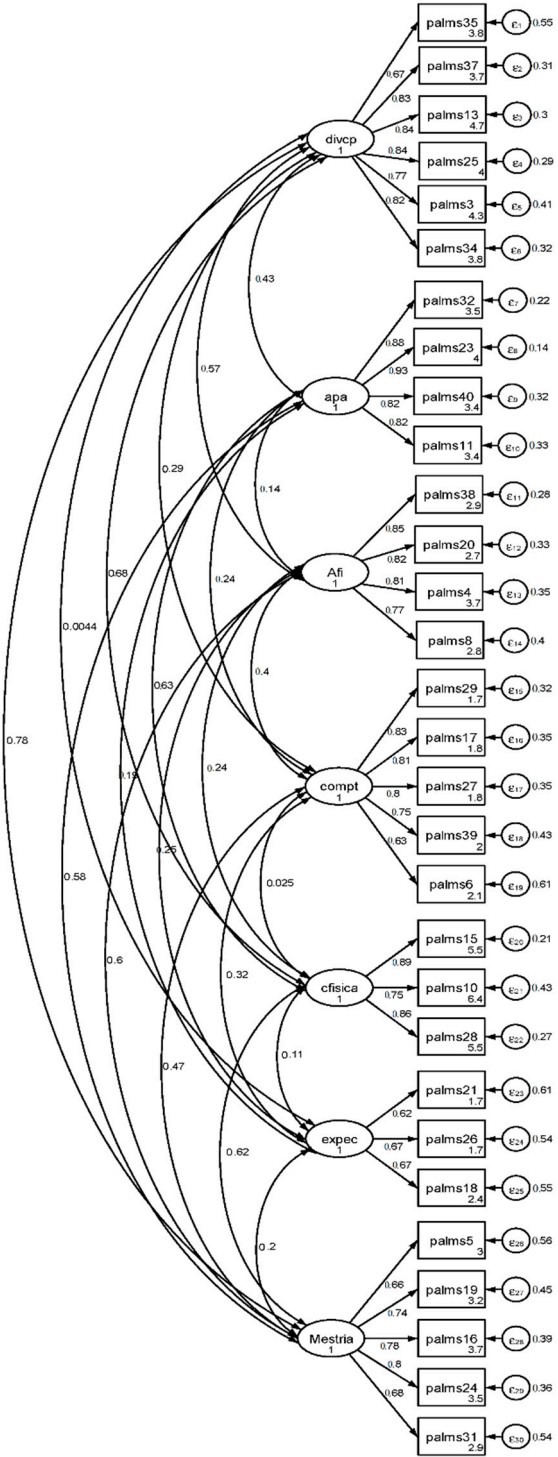

**Figure A1.** The PALMS-p measure model obtained through an exploratory approach, with elimination of items [$\chi^2$/df = 2.010, CFI = 0.950, GFI = 0.855, TLI = 0.939, RMSEA = 0.021, P(rmsea ≤ 0.05) < 0.001]. **Caption:** Divcp, Fun; Compt, Competition/Ego; Apa, Appearance; Expec, Expectation of Others; Afi, Affiliation; Cfísica, Physical Condition; Mestria, Mastery.

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
