# Peer review of "Validation of the Portuguese Adaptation of the Physical Activity and Leisure Motivation Scale (PALMS-p)"

_sustainability, doi:10.3390/su12145614_

Round 1
Reviewer 1 Report
This manuscript uses confirmatory factor analysis techniques to investigate the factorial validity of a Portuguese adaptation of the Physical Activity and Leisure Motivation Scale (PAMLS-p) instrument. The authors also investigate measurement invariance according to sex. The final confirmed model consisted of seven dimensions and 30 items. The authors also find that the factorial structure of the scale is stable between men and women in their sample. The topic is important and the methods appear to be sound. However, there are some issues which need to be addressed before the manuscript can be considered for publication. I hope the authors will find these comments useful for improving their manuscript.
Comments:
- My main concern is with the English language and grammar used throughout the manuscript. The manuscript needs to be thoroughly edited, preferably by someone who speaks English as a first language. There are several instances where the intended meaning is not entirely clear. Some examples include:
- The use of the word “adhesion” (i.e. “adhesion to the practice of PA”). Should this be “adherence”?
- Line 44-46: “…connected to the need to improve health and fitness, being the more frequently referred barriers lack of time and lack of motivation”. This is not grammatically correct.
- Typing errors: “physical” at line 95; “Cronbach” at line 100; “…as in, in no section…” at line 124.
- Convoluted language in places: “average questionnaire filling time”, should this be “average time to complete the questionnaire”? The first lines of the abstract (lines 21-24) and introduction (lines 37-39) are long and difficult to follow.
- Acronyms presented in the wrong order. E.g. “AFC” is used instead of “CFA” to denote “confirmatory factor analysis”; “IM” is used instead of “MI” to denote “modification index”.
- I would suggest reframing the title of the manuscript to better reflect that you are validating the Portuguese adaptation of the PALMS instrument. Suggested title: “Validation of the Portuguese Adaptation of the Physical Activity and Leisure Motivation Scale (PALMS-p)”.
- Introduction, lines 39-41: This line seems vague. Why is PA considered the cheapest instrument of public health? What are the cost savings? What are the benefits for QOL, health and wellbeing?
- There are several instance where a reference is cited in the middle of a sentence. For example, line 69: “In an attempt to rectify the limitations brought up above [17] created an instrument…”. It would be more usual to insert the author names and year here, and move the citation to the end of the sentence.
- Methods, lines 99-100: “In previous research this instrument presented high internal consistency values through (0,82)”. It is unclear what is meant by “(0,82)” here. Does this mean previous research has reported internal consistency values (Cronbach’s alpha) between 0 and 0.82? If so, would a Cronbach’s alpha value of 0 be considered to represent high internal consistency?
- Translation and adaptation: Did the adaptation follow any published formal framework? If so, can you insert the reference?
- Data analysis: What version of Stata was used? Can you insert the reference? E.g. for version 14: StataCorp: Stata Statistical Software: Release 14. College Station, TX: StataCorp LP. 2015.
- Data analysis: I think this section could benefit if clarification is provided on a few points:
- Chi-square test, ratio between chi-square and degrees of freedom, Comparative Fit Index, Tucker-Lewis Index, Root-Mean-Square Error of Approximation. Clarify that these are all goodness-of-fit indices. Provide a brief overview of what aspect of “model fit” each measure is assessing. E.g. the chi-square test examines absolute model fit, the Comparative Fit Index compares the chi-square statistic from the specified model with the chi-square statistic from the null model, in which all of the variables are uncorrelated.
- Clarify what criteria is used for the chi-square test and ratio between the chi-square and degrees of freedom tests to be considered as indicating good model fit.
- Define the terms discriminant validity and internal consistency. What values of extracted average variance and “composite reliability” are considered as indicating good discriminant validity and internal consistency?
- Results, lines 145-147: This does not appear to be part of the reporting of the results of the manuscript. It looks like it has been retained from a manuscript template.
- Results, refinement process: What is the rationale for removing items with factor loadings <0.50? Do you have a reference for this cut-point? Similarly, what is the “modification index criterion”? It might be useful to include a definition of “modification index” (reduction in the chi-square value when a specific parameter is freed)? Again, what is the rationale for using MI values above 50? This information should be moved to the methods section.
- Results: References to “factor load”, “factorial load” etc. I would keep the terminology consistent (“factor loading”?). Lines 163-165: keep the phrasing of results in the parentheses consistent.
- Results: The procedures for assessing measurement invariance need to be included in the methods section. Can you provide supporting statistics throughout the results section for the AVE values etc. What were the model fit indices obtained for your final 30-item model?
- Discussion section: Change “the modification of indexes” to “modification indices” at line 195. Line 204-207, sentence is poorly phrased and not grammatically correct. Change “unables the ability to make conclusions” to “means we are unable to make conclusions” at line 221.
- Table A5: I would suggest including the English version of the items side-by-side with the Portuguese. Also, this table has not been referenced in text. I would suggest that this should be labelled Table A2 and inserted at the "Translation and adaptation" subsection of the methods section.
Author Response
Dear Reviewer:
It is with great pleasure that we send the revised version of our manuscript. We believe that your comments and suggestion were of great importance to improve the quality of the present study. The attached document presents the modifications taking into account your comments.
Grateful for the attention.
Best regards,
João Lameiras- Portuguese Athletics Federation

Reviewer 2 Report
- This topic has some values for the Portuguese version of the Physical Activity and Leisure Motivation Scale. I suggest the authors can provide the importance of this scale, and why to advise this scale in Portugal.
- The authors employ the appropriate statistical method to deal with the sample data. I suggest the authors can provide some descriptions or explorations about the latent factors in this scale.
Author Response

(The authors gave the same response as above.)

Round 2
Reviewer 1 Report
The authors have been very responsive to my comments, and I think that the manuscript is improved. However, I have a few more comments for you to consider.
- There is still no reference to the statistical procedures for assessing measurement invariance in the methods section (it comes as a surprise in the results section). You should include a brief overview of the “parameter invariance technique” in the methods section and state your intention to use it to assess measurement invariance for men versus women. It would also be useful to provide an overview of the differences between Models A, B and C. For example, when you refer to “restricted factor loads” for model B, do you mean that in model B factor loadings were constrained to be equal for men and women? For model A, “total configuration invariance” needs defined. For model C, “complete scalar invariance” needs defined. All of this information needs added to the methods section.
- Can you check reference 36. I think the year should be 2014 instead of 2004?
- The numbering of the tables needs updated. Table A4 should be Table A3, and you have two tables named “Table A5”. I also think that your current Table A6 needs to be removed. Am I correct in assuming that this has been replaced by Table A2 in the current version of the manuscript? It is also not referred to in the manuscript text.
